

# Oral health manifestations and the perceived quality of life among Saudi children: a cross-sectional study

Heba Jafar Sabbagh[1] and Shahad N. Abudawood[2]

[1] Pediatric Dentistry Department, King Abdulaziz University Faculty of Dentistry, Jeddah, Saudi Arabia
[2] Pediatric Dentistry Department/Faculty of Dentistry, King Abdulaziz University, Jeddah, Saudi Arabia

## ABSTRACT

**Background:** This cross-sectional study addresses scarcity of evidence on oral health issues among Saudi children and their impact on quality of life (OHRQL). We aimed to investigate parental reports of oral health problems in children and their effect on their OHQRL.

**Methods:** Parents of children aged 2–11-years from Five-Saudi regions participated from February 2021 to July 2021 by completing an electronic, self-administered questionnaire structured according to World Health Organization-Oral Health Questionnaire for children. It comprises questions on children experiencing oral pain/discomfort, oral lesions/manifestations and reduction in their OHRQL.

**Results:** Among 1,516 responders, 1,107 (73.0%) reported that their children experienced toothache/discomfort. The possibility of parents reporting toothache/discomfort or oral manifestations decreased with younger children. For children aged 2–5 years, the odds ratio (AOR) was 0.18 (95% CI [0.13–0.24], $P < 0.001$) for toothache and 0.58 (95% CI [0.45–0.74], $P < 0.001$) for oral manifestations. For children aged 6–8 years, the AOR was 0.57 (95% CI [0.4–0.81], $P = 0.002$) for toothache. Additionally, parents of younger children less frequently reported reduced OHRQL with AORs of 0.58 (95% CI [0.45–0.73], $P < 0.001$) for children aged 2–5 years and 0.64 (95% CI [0.49–0.83], $P < 0.001$) for those aged 6–8 years. Lower parental education increased AORs, with values of 1.575 (95% CI [1.196–2.074], $P = 0.001$) and 1.505 (95% CI [1.208–1.876], $P < 0.001$) for younger and 6–8-year-old children, respectively.

**Conclusion:** Results revealed notable prevalence of toothache/discomfort and oral manifestations in children reported by parents, which was related to age and parental education; ultimately leading to reduction in their OHRQL.

## INTRODUCTION

Oral health-related quality of life (OHRQoL) is an essential component of overall health. It is referred to by the World Health Organization (WHO), as the impact of different oral health problems on individuals' physical, psychological, and social aspects of quality of life

Corresponding author
Heba Jafar Sabbagh,
hsabbagh@kau.edu.sa

and well-being (*Locker & Allen, 2007*; *Sischo & Broder, 2011*). Multiple oral conditions and poor oral health are associated with worse OHRQoL (*Mulla, 2021*). Dental caries and malocclusion can result in pain, reduced chewing capacity, difficulty eating, hypersensitivity to hot or cold beverages, decreased appetite, weight loss, and embarrassment, thus negatively affecting an individual's quality of life (*Edelstein & Reisine, 2015*; *Martins-Júnior et al., 2013*; *Ramos-Jorge et al., 2014*). Poor oral health in children has extensive negative effects on their growth, cognitive development, and daily activities (*Blumenshine et al., 2008*; *Sheiham, 2006*). This can diminish their learning ability, resulting in poor school attendance, educational performance, social interactions, and their overall quality of life (*Chen et al., 2020*; *Pakkhesal et al., 2021*; *Rebelo et al., 2018*).

Notably, the incidences of oral health and dental caries differ across socioeconomic groups. Previous studies have examined the effects of socioeconomic factors including parental age, sex, employment status, educational level, oral health literacy, and household income, on children's oral health (*Adil et al., 2020*; *Bozorgmehr, Hajizamani & Malek Mohammadi, 2013*). Available evidence suggests that highly educated parents have better oral health knowledge and are more receptive to preventive measures, such as fluoride applications and pit and fissure sealants. They also tend to regularly brush their children's teeth and visit the dentist. Consequently, children of highly educated parents tend to have better oral hygiene. Moreover, maternal educational level plays a crucial role in shaping their children's oral health knowledge and practices (*Dumitrescu et al., 2022*; *Ellakany et al., 2021*). Oral health literacy is a significant concern in relation to individuals' oral health. Different studies have suggested that low oral health literacy is associated with poor oral health knowledge (*Sabbahi et al., 2009*). Additionally, previous studies have demonstrated an association between low parental oral health literacy and high caries risk in children (*Adil et al., 2020*; *Neves Érick et al., 2020*).

While some studies have explored the relationship between oral health problems and oral health-related quality of life (OHRQoL) across various populations, few have specifically examined this issue among Saudi Arabian children (*Bamashmous et al., 2024*; *Barbosa & Gavião, 2008*). The healthcare system in Saudi Arabia has experienced significant improvements over the past decade, promoting preventive health measures and providing comprehensive services *via* specialized healthcare providers (*Almalki, Fitzgerald & Clark, 2011*). Despite these advancements, significant challenges remain, such as disparities in access to care, a high prevalence of dental caries and other oral health issues among children, and limited oral health awareness among parents. Such concerns remain critical in the government and community service sectors (*Almajed et al., 2024*; *Gaffar et al., 2022*; *Vundavalli & Baig, 2022*). Understanding the extent of the problem by evaluating parental reports of their children's oral problems and the OHRQoL is the first step in guiding future policies, governmental strategies, and community health initiatives.

Therefore, this cross-sectional study, conducted in Saudi Arabia, aimed to assess parental reports of children's oral health problems. The impact of oral health problems on children's OHRQoL in Saudi Arabia and the related sociodemographic factors was also investigated.

## MATERIALS AND METHODS

This cross-sectional study adhered to the Strengthening the Reporting of Observational Studies in Epidemiology (STROBE) guidelines. The questionnaire was distributed among parents of children, aged 2–11 years, throughout the different regions of the kingdom including the central (Riyadh, Qasim), western (Mecca, Medina, Jeddah), eastern (Damam, Khafi, Alhasa), northern (Tabuk, Jouf, Hail), and southern (Asir, Najran, Jizan) regions (*Farrag et al., 2019*). Research approval was obtained from the Research Ethics Committee of King Abdulaziz University, Faculty of Dentistry (Approval No. 232-03-21). The sample size was calculated based on the previously reported prevalence of pain in children (42.6%) (*Pentapati, Yeturu & Siddiq, 2021*). Using OpenEpi, it was estimated that 322 participants for a design effect of 2, a 95% confidence interval, and 80% power.

The inclusion criteria were parents of healthy, unaffected children, aged 2–11 years, living in Saudi Arabia. Parents of children who lived outside Saudi Arabia or children outside the specified age range were excluded. Data were collected between February 2021 and July 2021.

The questionnaire was structured according to the World Health Organization (WHO) oral health surveys and a basic methods questionnaire for children, 5th edition (2013) (*Farrag et al., 2019*; *World Health Organization, 2013*). This Basic oral health survey was conducted to evaluate the oral health status of the population and anticipate future dental care needs. It has a history of employing epidemiological survey methodologies including easily understandable diagnostic criteria that can be implemented in public health programs globally. This study includes parts I–III as follows.

### Part I

This section of the oral health survey collected general caregiver information including demographic data, parental education level, and region of residence. Parental education is also an important indicator of socioeconomic status (*Weinberg et al., 2019*). In Saudi Arabia, parental education is viewed as more significant than family income because government healthcare facilities, which maintain high standards, are free of charge for all Saudi citizens (*Gurajala, 2023*; *The Embassy of the Kingdom of Saudi Arabia, 2019*).

Additionally, Part I included general information on the children including demographic data, birth order, and dental history. Children were divided into three age groups: 2–5 years (toddlers and preschoolers with full primary dentition), 6–8 years (early elementary school-aged children with early mixed dentition), and 9–11 years (late elementary school-aged children with late mixed dentition) (*Dashash & Al-Jazar, 2018*). These age subgroups provided a more detailed understanding of the age-dependent needs of the children. The information has also been used to assess OHRQoL indicators. Furthermore, parental education was categorized as "high school or lower" and "higher education" based on the distribution of education in Saudi Arabia and the expected differences in various education categories (*Ministry of Education of Saudi Arabia, 2020*).

## Part II

This section described the children's oral health problems using two indicators. Indicator 1 comprised parental reports of children's oral pain/discomfort experiences during the past 12 months (categorized as often, occasionally, rarely, or never). Indicator 2 comprised parental reports of their children experiencing oral lesions or manifestations, such as pain while eating, tooth discoloration, bad odor/breath, swollen gums, abscess, dry mouth, tumor, difficulty speaking or eating, ulcers, white spots, skin roughness, or burning sensation during the past 12 months, (yes or no responses). These indicators are listed in Table 1 (El Tantawi et al., 2022).

## Part III

This section comprised parental reports of children's OHRQoL during the past 12 months. Indicator 3 consisted of parental reports of children's OHQoL (responses: yes, I do not know, or no) according to five items: "Were their children satisfied with their teeth appearance?" 'Did their children avoid smiling in public?' "Did other children make fun of their children's teeth?" 'Did their children experience difficulty biting hard food?', and "Did their children experience difficulty in chewing?".

The questionnaire was validated by five experts in English (content validity index (CVI) = 0.94) and Arabic (CVI = 0.98). For indicator 1, parents who reported that their children often, occasionally, or rarely experienced pain/discomfort were grouped and compared with those who reported that their children had never experienced pain/discomfort. For indicator 3, parents who responded "yes" or "I do not know" indicating suboptimal or uncertain optimal OHRQoL, were compared with those who responded "no" to any of the five questions indicating optimal OHRQoL.

### Method of questionnaire administration

We adopted a snowball sampling technique that applied multiple routes for distribution by assigning a data collector from each Saudi Region. The data collector distributed the self-administered questionnaire to parents *via* teachers, primary healthcare providers, and mothers to ensure that different Saudi population groups were reached. Electronic platforms, such as WhatsApp, Snapchat, Instagram, and Twitter, were used for survey distribution. Each data collector was based in a different region of Saudi Arabia. They began by reaching out to schoolteachers and healthcare providers from various schools and hospitals within their region. Additionally, they contacted mothers in different social circles, particularly those with extensive community connections.

Written consent to participate was displayed on the screen when participants accessed the questionnaire. Consent was obtained at the beginning of the electronic survey and respondents were required to agree before proceeding. Participation was voluntary and participants could exit the survey at any time for any reason before submitting their responses. If the parents had more than one child, they were asked to select the youngest child within the study age range to report on.

The data were tabulated and interpreted by one of the researchers (HJS) according to the reported categories described in the questionnaire structure and the WHO oral health

**Table 1 Parental report for their children's complaining of toothache/discomfort, detection of oral lesions/manifestations or suffering from reduction of their quality of life's due to any of the oral health-related problems.**

| Variable (parental report) | | Frequency (%) |
| --- | --- | --- |
| (A)-Parental report of their children complained of toothache toothache/discomfort | | |
| Have your child complained of toothache or feel discomfort due to teeth during the past 12 months? | Yes | 1,107 (73.0) |
| | No | 409 (27.0) |
| How often during the past 12 months did you have toothache or feel discomfort due to your teeth? | Often | 76 (5.0) |
| | Occasionally | 496 (32.7) |
| | Rarely | 535 (35.3) |
| | Never | 409 (27.0) |
| (B)-Parents detected any oral lesions/manifestations | | |
| Have you detected any oral lesions/manifestations in your child oral cavity in the last 12 months | Yes | 725 (49.5) |
| | No | 739 (50.5) |
| Pain while eating | Yes | 175 (11.5) |
| | No | 1,341 (88.5) |
| Teeth discoloration | Yes | 194 (12.8) |
| | No | 1,322 (87.2) |
| Bad odour/bad breath | Yes | 231 (15.2) |
| | No | 1,285 (84.8) |
| Swollen gums/swelling | Yes | 72 (4.7) |
| | No | 1,444 (95.3) |
| Abscess cyst | Yes | 53 (3.5) |
| | No | 1,463 (96.5) |
| Dry mouth | Yes | 22 (1.5) |
| | No | 1,494 (96.5) |
| Itchy gums | Yes | 14 (0.9) |
| | No | 1,502 (99.1) |
| Tumour | Yes | 12 (0.8) |
| | No | 1,504 (99.2) |
| Difficulty in speaking | Yes | 14 (0.9) |
| | No | 1,502 (99.1) |
| Ulcer/Multiple ulcers | Yes | 28 (1.8) |
| | No | 1,488 (98.2) |
| White spots in the mouth or gums | Yes | 85 (5.6) |
| | No | 1,431 (94.4) |
| Roughness in the skin of the mouth | Yes | 12 (0.8) |
| | No | 1,504 (99.2) |
| Burning sensation | Yes | 5 (0.3) |
| | No | 1,511 (99.7) |
| Others | Yes | 166 (10.9) |
| | No | 1,350 (98.1) |

(Continued)

| Table 1 (continued) | | |
|---|---|---|
| **Variable (parental report)** | | **Frequency (%)** |
| (C)-Parental report of their children suffering from reduction of their quality of life due to oral related problems | | |
| Have you experienced reduction of their quality of life due to teeth/mouth problem related to any of the following five-questions during the past 12 months | Yes/I do not know | 706 (46.6) |
| | No | 810 (53.4) |
| My child is not satisfied with the appearance of his teeth | Yes | 284 (22.8) |
| | No | 960 (77.2) |
| | I don't know | 272 (17.9) |
| Usually, my child avoids smiling in public | Yes | 112 (7.4) |
| | No | 1,301 (85.8) |
| | I don't know | 103 (6.8) |
| Are other children make fun of my child teeth | Yes | 84 (5.5) |
| | No | 1,331 (87.8) |
| | I don't know | 101 (6.7) |
| My child has difficulty biting hard food | Yes | 160 (10.6) |
| | No | 1,265 (83.4) |
| | I don't know | 91 (6.0) |
| My child has difficulty in chewing | Yes | 162 (10.7) |
| | No | 1,262 (83.2) |
| | I don't know | 92 (6.1) |

survey. Statistical analyses were performed using SPSS, version 23 (IBM Corp., Armonk, NY, USA). Categorical variables are presented as frequencies and percentages, and comparisons were made using the chi-squared test. Binary regression analysis was conducted three times for the three indicators of OHRQoL (dependent variables). Sex, age, geographic zone, and parental education were independent variables. Odds ratios were adjusted (AORs) to minimize the effects of confounders (Santos et al., 2008). Child order and the number of siblings were excluded from the regression analysis to avoid collinearity with the child's age. Paternal education was excluded to prevent collinearity with maternal education. Statistical significance was set at $P < 0.05$.

## RESULTS

Questionnaires were completed by 1,516 parents, with 463 (30.5%) residing in the western, 477 (31.5%) in the central, 263 (17.3%) in the southern, 182 (12.0%) in the eastern, and 131 (8.6%) in the northern regions. Respondents included 784 men (51.7%) and 732 women (48.3%). Additionally, 576 (38.0%) participants were parents of children, aged 2–5 years (Table 2).

Within our study cohort, 1,107 (73.0%) parents reported that their children experienced toothache or teeth-related discomfort. Additionally, 725 parents (49.5%) reported that their children had oral lesions or other oral manifestations. The most frequently reported

**Table 2 Participant's characteristics.**

| Variables | | Frequency (%) |
|---|---|---:|
| Region of residence | South | 263 (17.3) |
| | North | 131 (8.6) |
| | East | 182 (12.0) |
| | West | 463 (30.5) |
| | Central | 477 (31.5) |
| Child age | 2–5 | 576 (38.0) |
| | 6–8 | 410 (27.0) |
| | 9–11 | 530 (35.0) |
| Child order | 1st child | 290 (19.1) |
| | 2nd & 3rd | 426 (28.1) |
| | ≥4th | 617 (40.7) |
| | Only child | 183 (12.1) |
| Gender | Male | 784 (51.7) |
| | Female | 732 (48.3) |
| Maternal education | ≤High school | 489 (32.3) |
| | ≥University | 1,027 (67.7) |
| Paternal education | ≤High school | 516 (34.0) |
| | ≥University | 1,000 (66.0) |

manifestations were pain while eating (175 (11.5%)), tooth discoloration (194 (12.8%)), and bad odor or breath (231 (15.25%)). Furthermore, 706 (46.6%) parents reported that their children had a reduced OHRQoL due to tooth/mouth problems. Importantly, 284 (22.8%) parents attributed OHRQoL to tooth appearance (Table 2).

Table 3 presents a comparative analysis of parental reports of their children's oral health problems and oral health-related quality of life in relation to sociodemographic factors. Child sex, birth order, and parental education level showed significant relationships with child oral health indicators ($P < 0.05$). Table 4 presents a regression analysis of parental reports of their children's oral health problems, OHRQoL (dependent factors), and the associated sociodemographic predictors. The odds ratio (OR) was adjusted to account for the influence of other predictors in the model, thereby minimizing the impact of confounding variables (Santos et al., 2008). Paternal education and child order were not included in the regression analysis to avoid collinearity with maternal education and child age. Compared with the parents of older children, parents of younger children less frequently reported that their children complained of toothache/discomfort or oral lesions/manifestations, with a decreased adjusted OR (AOR) (for toothache/discomfort: children aged 2–5 years (AOR: 0.18, 95% CI [0.13–0.24] and $P < 0.001$) and children aged 6–8 years (AOR: 0.57, 95% CI [0.4–0.81] and $P = 0.002$); and for oral manifestations: children aged 2–5 years (AOR: 0.58, 95% CI [0.45–0.74] and $P < 0.001$)). Compared with the parents of girls, parents of boys less frequently reported that their children complained of toothache/discomfort and a decreased AOR (AOR: 0.66; 95% CI [0.52–0.84] and $P < 0.001$).
**Table 3 Distribution of the sample according to parental reports of their children's oral health problem (indicator 1 and 2) and their reduction of quality of life due to oral health-related problems, according to sociodemographic factors.**

| Variables | | Indicator 1: complained of toothache/discomfort | | | Indicator 2: detected any oral lesions/manifestations | | | Reduction of children quality of life due to oral related problems | | |
|---|---|---|---|---|---|---|---|---|---|---|
| | | Yes (%) | No (%) | P value[C] | Yes (%) | No (%) | P value[C] | Yes (%) | No (%) | P value[C] |
| Gender | Male | 549 (70.0) | 235 (30.0) | 0.007* | 372 (49.5) | 380 (50.5) | 0.966 | 364 (46.4) | 420 (53.6) | 0.909 |
| | Female | 558 (76.2) | 174 (23.8) | | 353 (49.6) | 359 (50.4) | | 342 (46.7) | 390 (53.3) | |
| Region | South | 201 (76.4) | 62 (23.6) | 0.053 | 129 (50.0) | 129 (50.0) | <0.001* | 133 (50.6) | 130 (49.4) | 0.648 |
| | North | 104 (79.4) | 27 (20.6) | | 65 (52.4) | 59 (47.6) | | 62 (47.3) | 69 (52.7) | |
| | East | 125 (68.7) | 57 (31.3) | | 73 (40.6) | 107 (59.4) | | 86 (47.3) | 96 (52.7) | |
| | West | 345 (74.5) | 118 (25.5) | | 257 (59.9) | 172 (40.1) | | 208 (44.9) | 255 (55.1) | |
| | Central | 332 (69.6) | 145 (30.4) | | 201 (42.5) | 272 (57.5) | | 217 (45.5) | 260 (54.5) | |
| Child order | 1st child | 189 (65.2) | 101 (34.8) | <0.001* | 136 (48.4) | 145 (51.6) | 0.001* | 109 (37.6) | 181 (62.4) | <0.001* |
| | 2nd & 3rd | 330 (77.5) | 96 (22.5) | | 190 (47.0) | 214 (53.0) | | 207 (48.6) | 219 (51.4) | |
| | ≥4th | 491 (79.6) | 126 (20.4) | | 328 (54.9) | 269 (45.1) | | 315 (51.1) | 302 (48.9) | |
| | Only child | 97 (53.0) | 86 (47.0) | | 71 (39.0) | 111 (61.0) | | 75 (41.0) | 108 (59.0) | |
| Child age (years) | 2–5 | 318 (55.2) | 258 (44.8) | <0.001* | 225 (40.9) | 325 (44.0) | <0.001* | 234 (40.6) | 342 (59.4) | <0.001* |
| | 6–8 | 326 (79.5) | 84 (20.5) | | 215 (54.2) | 182 (45.8) | | 178 (43.4) | 232 (56.6) | |
| | 9–11 | 463 (87.4) | 67 (12.6) | | 285 (55.1) | 232 (44.9) | | 294 (55.5) | 236 (44.5) | |
| Maternal education | ≤High school | 393 (80.4) | 96 (19.6) | <0.001* | 243 (52.1) | 223 (47.9) | 0.170 | 266 (54.4) | 223 (45.6) | <0.001* |
| | ≥University | 714 (69.5) | 313 (30.5) | | 482 (48.3) | 516 (51.7) | | 440 (42.8) | 587 (57.2) | |
| Paternal education | ≤High school | 407 (78.9) | 109 (21.1) | <0.001* | 271 (54.4) | 227 (45.6) | 0.007* | 276 (53.5) | 240 (46.5) | <0.001* |
| | ≥University | 700 (70.0) | 300 (30.0) | | 454 (47.0) | 512 (53.0) | | 430 (43.0) | 570 (57.0) | |

Notes:
[C] Chi square test.
* P value significant at 0.05.

**Table 4 Binary regression analysis for parental reports of their children's oral health problem and their oral health-related quality of life (dependent factor) and the associated sociodemographic factors.**

| Factors | | Indicator 1: toothache/discomfort AOR (95% CI) P value | Indicator 2: Oral lesions/manifestations AOR (95% CI) P value | Reduction of children quality of life due to oral related problems AOR (95% CI) P value |
|---|---|---|---|---|
| Gender | Male | 0.66 [0.52–0.84] <0.001* | 0.97 [0.79–1.2] 0.8 | 0.98 [0.80–1.21] 0.88 |
| | Female | 1.00 | 1.00 | 1.00 |
| Region | South | 1.44 [0.99–2.08] 0.06 | 1.35 [0.99–1.83] 0.06 | 1.21 [0.89–1.64] 0.23 |
| | North | 1.66 [1.01–2.72] 0.04* | 1.48 [0.99–2.22] 0.05* | 1.05 [0.71–1.56] 0.80 |
| | East | 1.05 [0.70–1.55] 0.83 | 0.95 [0.67–1.35] 0.77 | 1.133 [0.800–1.605] 0.480 |
| | West | 1.34 [0.99–1.82] 0.06 | 2.04 [1.56–2.66] <0.001* | 1.02 [0.78–1.32] 0.90 |
| | Central | 1.00 | 1.00 | 1.00 |
| Child age (years) | 2–5 | 0.18 [0.13–0.24] <0.001* | 0.58 [0.45–0.74] <0.001* | 0.58 [0.45–0.73] <0.001* |
| | 6–8 | 0.57 [0.4–0.81] 0.002* | 0.95 [0.73–1.24] 0.72 | 0.64 [0.49–0.83] <0.001* |
| | 9–11 | 1.00 | 1.00 | 1.00 |
| Maternal education | ≤High school | 1.58 [1.17–2.07] 0.001* | 1.15 [0.91–1.44] 0.24 | 1.51 [1.21–1.88] <0.001* |
| | ≥University | 1.00 | 1.00 | 1.00 |

Notes:
AOR, Adjusted odds ratio.
* P value significant at 0.05.

Compared with parents of older children, parents of younger children less frequently reported reduced OHRQoL due to oral problems, with a decreased AOR (AOR: 0.58, 95% CI [0.45–0.73] and $P < 0.001$; and AOR: 0.64, 95% CI [0.49–0.83] and $P < 0.001$, respectively). Moreover, compared with highly educated parents, those with a low lower education level more frequently reported that their children experienced teeth or mouth problems and complained of reduced quality of life due to oral problems, with an increased AOR (AOR: 1.58, 95% CI [1.17–2.07] and $P = 0.001$; and AOR: 1.51, 95% CI [1.21–1.88] $P < 0.001$, respectively).

## DISCUSSION

Our findings indicate that parents of Saudi children reported a high frequency of toothaches and oral health problems, which affected their children's OHRQoL. Previous studies have described the barriers that could interfere with ideal oral healthcare (*Sabbagh et al., 2022b*). Therefore, it is important to assess the prevalence of oral health problems and their effects on the OHRQoL of children with such problems.

Saudi Arabia's healthcare system is divided into two main sectors. The public healthcare system is managed by the Ministry of Health (MOH) and provides approximately 60% of the free healthcare services covering all regions in the country (*Ministry of Health of Saudi Arabia, 2014*) and the private sector provides approximately 23% of the services. Describing the different regions in the analysis offered insights into the diversity of the country's culture (*Long, 2005*) and allowed us to explore the differences reported in previous studies. These differences include accessibility to public dental clinics and distribution of health care providers (*Alqahtani et al., 2022*), the country's variation in children caries experience (*Adam et al., 2022*), socioeconomic status, and population size which could act as confounders, potentially influencing our findings on oral health care quality of life (*Adli, 2011*; *Al Agili & Farsi, 2020*; *Allaf et al., 2022*; *Bahannan et al., 2018*; *Global Media Insight Research Team, 2022*; *Kannan et al., 2020*; *Pimentel et al., 2014*). For example, the central region, which encloses the capital city, is expected to have a higher population and services (*Al-gabbani, 1992*). *Shubayr, Kruger & Tennant (2021)* assessed the accessibility of public dental services in the Jazan area in 2022. They found that only 31% of the Jazan residents lived in a serviced area (*Shubayr, Kruger & Tennant, 2021*). More research is needed to assess the accessibility of public dental clinics across the kingdom.

A high prevalence of dental pain over the past 12 months was reported in our study (approximately 75%). The prevalence of reported pain in the southern and western regions was 54% (*Kannan et al., 2020*) and 44.6%, respectively (*Bahannan et al., 2018*). Regarding the worldwide prevalence, *Pentapati, Yeturu & Siddiq (2021)* recently reported that the prevalence of dental pain among children and adolescents ranged from 1.33% to 87.8%. The highest and lowest prevalence rates of dental pain were reported in Africa (50.1%) and Australia (20.7%), respectively. A high prevalence of dental pain has been reported in India (40.4%), China (41.3%), and Iran (42.6%) (*Pentapati, Yeturu & Siddiq, 2021*). The differences in the reported prevalence rates of dental pain could be attributed to among-study differences in the sampling technique, inclusion criteria, children's age

group, sample size, pain assessment surveys/methods, geographic location, and cultural differences (*Im et al., 2007*; *Plesh, Adams & Gansky, 2011*; *Tay et al., 2021*). However, our study reported the highest prevalence rates among the currently reported values, indicating the need for the prompt implementation of oral healthcare measures. This could also be attributed to the fact that this study was conducted during the COVID-19 pandemic, which may have affected patients' ability to access health services and dental care (*Farook et al., 2020*; *Ibrahim et al., 2021*; *Sabbagh et al., 2022b*). It is well documented that the novel coronavirus disease (COVID-19) outbreak has affected the access to dental and oral health services in many nations worldwide (*Odeh et al., 2020*; *Torlińska-Walkowiak et al., 2023*). In Saudi Arabia, regulatory entities have imposed regulations to close dental clinics and halt routine dental treatments, except for those who need emergency care, to limit viral transmission (*Odeh et al., 2020*). *Alamoudi et al. (2022)*, reported a reduction in utilization of dental services by children during the pandemic period at a King Abdul Aziz University Hospital in Saudi Arabia compared to the same period in the previous year before the pandemic. This might have affected the prevalence of reported oral health problems and the results of this study (*Danagoulian & Wilk, 2022*).

Regarding parent-reported oral manifestations/lesions, half of the parents reported bad odors/bad breath and tooth discoloration as the most common oral manifestations. This finding is consistent with previous reports of a high prevalence of parent-reported oral manifestations, including halitosis, in children (*Rosenberg, Amir & Robinson, 2002*; *Sabbagh et al., 2022a*). In addition to its negative social effects, halitosis is one of the most common symptoms of oral pathologies. In up to 85% of the cases, halitosis originates from the oral cavity and/or an otorhinolaryngological source, such as sinus-related illness and tonsillitis. Oral halitosis is related to poor oral hygiene, dental caries, plaque, and calculus build-up, leading to gingival and periodontal problems and providing favorable conditions for bacterial colonization. However, in the remaining 15% of the cases, the source was non-oral halitosis, which could be related to a systemic disease. Oral halitosis is usually treated by the mechanical/chemical reduction of intraoral microorganisms (*Nakhleh, Quatredeniers & Haick, 2018*). However, dentists may be the first to detect halitosis and make appropriate referrals if nonoral halitosis is diagnosed. Based on our results, we would suggest evaluating pediatric patients during regular dental visits for halitosis to determine the cause and rule out non-oral causes.

Furthermore, half of the parents reported that their children had a reduced quality of life due to teeth/mouth problems. Specifically, 22.8%, 10.7%, 10.6%, and 7.4% of the children were unsatisfied with their teeth appearance, had difficulty chewing hard food, and avoided smiling publicly, respectively. Consistent with our findings, *Clementino et al. (2015)* reported a high rate of parent-reported difficulties in eating food and avoidance of smiling by their children. This demonstrates the importance of oral healthcare for children.

In addition, parents of boys reported that their children complained of toothache/discomfort more frequently than parents of girls. Compared with males, it has been reported that females tend to have a higher risk and prevalence of caries (*Ferraro & Vieira, 2010*; *Lukacs & Largaespada, 2006*). Furthermore, females are more sensitive to pain than males and demographic factors influence emotional expression in children

(*Chaplin, Cole & Zahn-Waxler, 2005*; *Fillingim, Edwards & Powell, 1999*; *Sorge & Totsch, 2017*). However, emotional expression in children varies with age.

Child age is a key indicator of oral healthcare practices, outcomes, and dental development (*Krol & Whelan, 2023*; *Moca et al., 2021*). Accordingly, we categorized the participants into subgroups based on age to represent the different stages of dental development. Younger children are expected to have fewer oral manifestations and dental problems than older children, because of their relatively shorter duration of exposure to teeth in the oral cavity. However, it is important to consider that younger children may lack the communicative ability and language to clearly express their dental discomfort or pain. Moreover, younger children may depend on their parents for pain recognition. Toddlers with oral problems may become inactive and avoid interacting with other children, or they may become very active, irritable, or crying. Other non-verbal pain expressions include changes in appetite and sleep patterns (*Mathews, 2011*).

However, the association between parental awareness of a child's oral health status and birth order remains unclear. Although the siblings share the same parents and households, they have different health issues and dental experiences. In preschool children, dental caries have been significantly associated with family size, childbirth rank, and parents' age at childbirth. Our results are consistent with those of *Julihn et al. (2020)* and *Wellappuli & Amarasena (2012)*, indicating that birth rank/order is a risk factor for developing caries in young children. In both previous studies, later-born children had a significantly increased risk of developing carious lesions compared with first-born children.

In our study, parents with lower educational levels reported that their children experienced tooth or mouth problems more frequently than did parents with higher educational levels. Parental knowledge and attitudes toward oral hygiene and health are crucial, and significantly reduce the prevalence of dental caries in children (*Dumitrescu et al., 2022*). *Ellakany et al. (2021)* reported that socioeconomic status, including parental education and family income, was related to familial lifestyle and eating habits, which influenced the prevalence of dental caries in children. Our findings are consistent with a report by *Ellakany et al. (2021)* that children of parents with higher education levels have a significantly lower prevalence of decayed teeth. Additionally, *Chen et al. (2020)* reported that highly educated parents have more knowledge about oral health than less-educated parents; therefore, their children have better oral hygiene. Additionally, they reported that toothbrushing behavior and frequency among children can be predicted by their parents' level of education, especially that of their mothers (*Chen et al., 2020*). Children of mothers with college/university education or higher were more likely to brush their teeth twice daily, brush more often, and visit the dental clinic regularly than children of mothers with middle school education or lower (*Sabbagh et al., 2019*). Moreover, studies have examined the relationship between parents' educational level, employment status, and their children's oral health. One study revealed that both parental unemployment and lower educational levels were linked to poorer oral health status (*Bamashmous et al., 2024*; *Minervini et al., 2023*; *Sabbagh et al., 2019*; *Sabbagh & Alzain, 2024*).

Although our questionnaire was widely distributed to participants across various geographic regions in the country, the results of our study may not represent the

population because of its cross-sectional design. Nevertheless, the main limitation stemmed from the potential reporting bias by parents who answered the questions on behalf of their children and the subjective nature of reporting their children's oral health status, as they may have overestimated or underestimated their children's symptoms. However, the quality of life of children reported by their parents can be reliable (*Khin Hla et al., 2014*). An additional limitation is that the study design did not allow for the establishment of a causal relationship. Furthermore, the use of snowball sampling may have introduced selection bias. The participants were likely to share similar characteristics or know each other, which means that not every member of the population had an equal chance of being included in the sample. This introduces sampling bias, which limits the generalizability of the findings and prevents an accurate evaluation of the response rate (*Heckathorn, 2011*).

This study suggests the need for more research to assess the current state of oral health care and its impact on the quality of life, specifically in relation to dental care, family dynamics, and teledentistry, Comparing the outcomes with this study will help gauge the improvements in recent years, particularly post-resolution of the COVID-19 pandemic. Additionally, conducting community programs to evaluate and compare their effects on children's oral problems and quality of life through a longitudinal study is an important step that must be planned.

Furthermore, despite national initiatives to promote oral hygiene and dental awareness among children and their parents (*Elsadek & Baker, 2023*; *Gaffar et al., 2022*; *Vundavalli & Baig, 2022*), there remains a significant need to enhance awareness and practices related to oral health and their impact on children's overall development and quality of life. Addressing the oral health disparities identified in this study is crucial. This can be achieved through the establishment of dental homes; implementation of public dental health policies; and integration of oral health programs focused on screening, education, and care in government schools, targeting children from diverse socioeconomic backgrounds. In addition, clinicians must understand the importance of targeted oral health interventions and suggest tailored programs to address the specific needs of children based on the age and educational background of the parents.

## CONCLUSIONS

Our findings revealed a notable prevalence of oral pain/discomfort and oral lesions/manifestations in children as reported by their parents, which were related to their age and parental education, ultimately leading to a reduction in their oral health-related quality of life. Given these findings, we recommend the development and implementation of targeted oral health interventions for children in Saudi Arabia, emphasizing the significance of family structure and parental education.

### Funding

The authors received no funding for this work.

## Competing Interests

The authors declare that they have no competing interests.

## Author Contributions

- Heba Jafar Sabbagh conceived and designed the experiments, performed the experiments, analyzed the data, prepared figures and/or tables, authored or reviewed drafts of the article, and approved the final draft.
- Shahad N. Abudawood performed the experiments, authored or reviewed drafts of the article, and approved the final draft.

## Ethics

The following information was supplied relating to ethical approvals (*i.e.*, approving body and any reference numbers):

Ethical approval was obtained from the Research Ethics Committee of King Abdulaziz University Faculty of Dentistry (Approval No. 232-03-21).

## Data Availability

The raw data and codes are available in the Supplemental Files.

## Supplemental Information

Supplemental information for this article can be found online at http://dx.doi.org/10.7717/peerj.18556#supplemental-information.

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
