# Peer review of "Oral health manifestations and the perceived quality of life among Saudi children: a cross-sectional study"

_PeerJ, doi:10.7717/peerj.18556_

## Round 0.1 · original submission · Major Revisions

Dear authors,

Thank you for submitting your manuscript. While the study offers valuable insights into oral health disparities among Saudi children, the reviewers have identified several areas requiring significant revision. These include clarifying the methodology, particularly the sampling process and the administration of the electronic questionnaire, as well as enhancing the discussion on the study's limitations, potential clinical implications, and relevance to policy recommendations. Additionally, grammatical improvements and consistency in following the study's objectives throughout the manuscript are necessary. We encourage you to address these concerns thoroughly to strengthen the manuscript before resubmission.

Please address these concerns comprehensively to improve the manuscript’s quality and readability before resubmission. Additionally, ensure that all reviewer questions and comments are fully addressed to avoid multiple rounds of revisions.

Reviewer 1 ·

Basic reporting

General Comments:
This manuscript offers valuable insights into oral health disparities among Saudi children, with a particular focus on the role of parental education and geographic factors. The discussion provides a thorough comparison with other relevant studies and effectively highlights the study’s strengths and limitations.

- A more detailed exploration of how the COVID-19 pandemic may have influenced the prevalence of reported dental issues would be beneficial. The authors briefly mention this but do not provide substantial analysis or data to support the claim.
- It would be interesting to see more discussion on the potential interventions that could mitigate the oral health disparities observed in the study, particularly in terms of policy recommendations for improving access to dental care for lower socioeconomic groups.

Experimental design

Experimental Design:
The study fits well within the journal’s scope as it provides valuable insights into oral health problems and their impact on quality of life in Saudi children. The research question is relevant and timely, addressing a gap in the literature about oral health-related quality of life (OHRQoL) among Saudi children.
The cross-sectional design is appropriate, and the data collection through a structured WHO-based questionnaire is suitable for the study’s objectives. Ethical approval is appropriately mentioned, and informed consent procedures are clearly outlined.

However, there are some concerns regarding the methodology:
Sampling method: The snowball sampling technique may have introduced selection bias. It would be helpful if the authors could acknowledge this limitation more clearly in the discussion.
Age group stratification: The stratification of children into three age groups is logical, but a rationale for these specific age ranges could be provided. This would help readers understand how the age groups were determined and whether they align with developmental stages relevant to oral health outcomes.
Replication potential:The methods section is mostly detailed enough to allow replication, though providing additional specifics about the electronic questionnaire distribution process (such as response rate per platform) could enhance clarity.

Validity of the Findings
The findings are well-supported by the data presented, and the authors have made efforts to control for confounding variables. The statistical analyses appear sound, with appropriate use of binary logistic regression to explore the associations between sociodemographic factors and oral health outcomes.

- As a cross-sectional study, it is limited in its ability to infer causality. The authors should be cautious in their language when discussing the implications of the findings and avoid over-generalizing the results.
- The subjective nature of parental reports on children’s oral health status could introduce bias, as parents may either overestimate or underestimate their children’s symptoms.

Validity of the findings

Suggestions for improvement:
- Consider elaborating on the justification for choosing the specific regions for the study, providing more context about regional differences in oral health care accessibility in Saudi Arabia.
- It would also be helpful to provide a more detailed discussion of the statistical methods used, especially regarding the adjusted odds ratios (AORs) mentioned.

Suggested citations: The authors should consider adding the following relevant citations to strengthen the discussion regarding parental education and oral health outcomes:
1. Minervini, G., Franco, R., Marrapodi, M.M., ... Cervino, G., Cicciù, M. (2023). *The association between parent education level, oral health, and oral-related sleep disturbance: An observational cross-sectional study.* European Journal of Paediatric Dentistry, 24(3), pp. 218–223.
2. Contaldo, M., della Vella, F., Raimondo, E., ... Sinescu, C., Serpico, R. (2020). *Early Childhood Oral Health Impact Scale (ECOHIS): Literature review and Italian validation.* International Journal of Dental Hygiene, 18(4), pp. 396–402.

Additional comments

The manuscript is clearly written and adheres to the journal’s standards in terms of language, structure, and relevance to the field. The introduction provides adequate background information and highlights the importance of the study. The literature is well-referenced, drawing on relevant studies. However, the manuscript could benefit from some minor grammatical revisions, particularly in the Methods section, to improve clarity and flow.
The figures and tables presented are of high quality, well-labeled, and appropriately referenced in the text. The inclusion of raw data is in accordance with PeerJ’s policies and adds to the manuscript’s transparency.

·

Basic reporting

Dear Authors,

I read the current manuscript with great interest. Titled "Oral health manifestations and perceived quality of life among Saudi children: A cross-sectional study ”. The result of the current study is entirely predictable. The result doesn't contribute to clinical dentistry and doesn’t contribute to improving the oral condition of the patients. The manuscript needs corrections; please find below the comments on how to improve it.

Abstract:
1. Methods: 26 and 27, “an electronic, self-administered questionnaire” – did the authors use Google form? How was it distributed? kindly elaborate.
2. Kindly provide more MeSH keywords so the article is more accessible to search for after publication.

Introduction:
1. Highly educated parents have better oral health knowledge and are more receptive to preventive measures, such as fluoride application and pit and fissure sealants: what about the parent’s literacy concerning caries experience? kindly write briefly, find the below article for reference and cite accordingly
https://www.mdpi.com/2227-9067/7/8/101
2. Kindly write briefly on the problem statement for the current study.
3. Kindly present the background with the recent literature.
4. Kindly write about the questionnaire used in the current study, including recent studies

Experimental design

Method:
1. How did the authors calculate the sample size? Kindly explain.
2. an electronic, self-administered questionnaire” – kindly describe how the authors administered the questionnaire and participant consent.
3. As the authors mentioned, they adopted the “snowball sampling technique.” How did the initial sample selection conducted?
4. Kindly provide the data sheet and the questionnaire as a supplementary file.
5. Kindly mention the inclusion and exclusion criteria.
6. Why did the authors record parents' education level in state socioeconomic conditions? Isn’t it more relevant? Many studies showed the correlation between patients' oral and parents' socioeconomic conditions. Or you could have included both.

Results:
1. The results have been well presented.
2. The characteristic table covered relevant details.

Validity of the findings

Discussion:
1. The outcome of the study is well discussed.
2. Kindly write all the limitations of the current study.
3. Kindly write about clinical implementation or how the current study contributes to the clinical field.
4. Does this study help parents improve their kids’ condition, as most have dental problems?
5. Kindly discuss further research ideas.

Additional comments

Additional comments:
1. Requires English and grammar editing
2. Kindly follow the objectives of the current study from the introduction until the conclusion to be in the same flow, which will make the reader easy

---

## Round 0.2 · accepted · Accept

The authors have thoroughly addressed all reviewer comments and suggestions, resulting in a significant improvement in the manuscript. The revised version demonstrates a clear and well-executed study, with all methodological and reporting concerns effectively resolved. Therefore, I am pleased to recommend this manuscript for publication in its current form. Congratulations to the authors, and best wishes for their future research endeavors.

·

Basic reporting

no comment

Experimental design

no comment

Validity of the findings

no comment

Additional comments

Dear Authors,
The authors have addressed all the comments and suggestions reviewers gave, and the manuscript has dramatically improved. The manuscript can be accepted for publication in its current form. I would like to congratulate the authors and wish them all the very best in their future endeavours.

Best regards and keep well